# The Impact of Nutritional Therapy in the Management of Overweight/Obese PCOS Patient Candidates for IVF

**DOI:** 10.3390/nu15204444

**Published:** 2023-10-20

**Authors:** Caterina Meneghini, Claudia Bianco, Francesco Galanti, Valentina Tamburelli, Alessandro Dal Lago, Emanuele Licata, Mariagrazia Gallo, Cristina Fabiani, Roberta Corno, Donatella Miriello, Rocco Rago

**Affiliations:** 1Physiopathology of Reproduction and Andrology Unit, Sandro Pertini Hospital, Via dei Monti Tiburtini 385/389, 00157 Rome, Italy; francescogalanti@hotmail.it (F.G.); alessandro.dallago@aslroma2.it (A.D.L.); emanuele.licata@aslroma2.it (E.L.); mariagrazia.gallo@aslroma2.it (M.G.); cristina.fabiani@aslroma2.it (C.F.); roberta.corno@aslroma2.it (R.C.); donatella.miriello@aslroma2.it (D.M.); rocco.rago@aslroma2.it (R.R.); 2Department of Science, University “Roma Tre”, 00146 Rome, Italy; cla.bianco@stud.uniroma3.it; 3Independent Researcher, 00100 Rome, Italy; valentinatamburelli@gmail.com

**Keywords:** VLCKD, Mediterranean diet, PCOS, IVF, overweight

## Abstract

Polycystic ovary syndrome (PCOS) is the most common female endocrine disorder, and it has two main pathological aspects: reproductive and metabolic. Overweight/obesity is a risk factor in terms of adverse effects during hormone stimulation, a reduced response to ovulation induction regimens, reduced success of IVF, and an increased risk of obstetric complications. To resolve this vicious cycle of pathological events, weight loss and lifestyle modifications are promising strategies. Among these possible approaches, the consumption of a very-low-calorie ketogenic diet (VLCKD) or Mediterranean diet (MD) represents a valid option. In our study, 84 obese/overweight PCOS patients were recruited to evaluate the effects induced by the VLCKD and MD on weight, hormonal, and metabolic parameters. BMI decreased significantly among the VLKCD patients compared to the MD patients (both presenting *p* values < 0.0001 at 90 and 120 days), and a significant reduction in body circumference was observed. At the same time, HOMA index values statistically decreased for the VLCKD patients compared to those on the MD (*p* value < 0.001 at 90 days and *p* value < 0.05 at 120 days), and this phenomenon was also observed for AFC at 90 and 120 days (both *p* values < 0.001) and AMH at 90 days (*p* value < 0.05). Interestingly, the ovarian hyperstimulation syndrome (OHSS) incidence was statistically lower in the VLKCD patients compared to the MD patients (*p* < 0.001). We state that these dietary regimes may improve anthropometric parameters (such as BMI) and women’s reproductive health, restore menstrual regularity, and reduce the risk of OHSS. Regarding the different nutritional therapies, the results suggest that the VLCKD is an optimal choice for entry into IVF, especially in terms of the time range in which these results are achieved.

## 1. Introduction

Infertility is a prevalent feature of polycystic ovary syndrome (PCOS). In 75% of these women, it occurs due to oligo-ovulation or anovulation [1,2,3]. PCOS is often associated with an increased prevalence of metabolic syndrome, cardiovascular diseases, and type 2 diabetes [4]. A common sign of PCOS, which is not included in the diagnostic criteria, is insulin resistance (IR): women with PCOS have increased visceral and subcutaneous body fat due to higher androgen levels, so obesity plays a significant role in explaining the metabolic characteristics of PCOS [3]. These patients present elevated levels of low-density lipoprotein (LDL), triglycerides, and cholesterol and reduced levels of high-density lipoprotein (HDL).

The maintenance of reproductive function is related to an optimal body weight in women: underweight as well as overweight and obesity are associated with an increased risk of anovulatory infertility. During ovarian stimulation, the FSH requirement is increased, fewer oocytes are collected, serum estradiol concentrations are decreased, frequent cycles are cancelled, and low pregnancy rates have been observed. The treatment of infertility caused by PCOS includes lifestyle changes, pharmacological and surgical therapies, and in vitro fertilization (IVF) [5,6,7,8,9]. It has also been reported that obese women are at an increased risk of miscarriage after IVF or ICSI.

The reduced fertility in these patients may also be related to other factors, such as reduced oocyte and/or embryo quality, defects in endometrial development, and implantation abnormalities. Furthermore, PCOS may have significant implications for pregnancy outcomes and the long-term health of PCOS-affected women and their offspring [10].

IVF has become a viable alternative for women with PCOS when ovulation induction treatment has failed or there are coexisting tubal or male infertility factors.

Given the harmful effect of obesity, weight loss has been strongly promoted as one of the most effective means of increasing fertility in infertile overweight or obese women and improving the outcomes of assisted reproductive treatment among obese women.

Ovarian stimulation in PCOS patients is a crucial and challenging process since the ovaries of such patients generally exhibit an excessive and uncontrolled response to hormonal stimulation. OHSS is the main risk for PCOS patients undergoing IVF; in a review, an odds ratio of 6.8 was calculated for the development of OHSS in association with polycystic ovarian morphology [11,12].

Although OHSS is relatively rare, women with this iatrogenic disease should be monitored carefully to avoid life-threatening complications. The evaluation of risk factors, such as PCOS and a high antral follicle count, is important [13]. Therefore, to reduce the risks, it is important to improve pregravid, prenatal, and intrapartum care. Approximately 60% of PCOS patients are overweight/obese (classification ranges are described in Table 1), and weight gain often precedes the onset of PCOS’ clinical manifestations. This suggests that obesity (especially the abdominal type) interferes with the pathophysiology and the clinical expression of the syndrome; the mechanisms underlying this association are complex and not completely understood. It has been observed that women with central obesity take longer to become pregnant and have more impaired ovulation, qualitative and quantitative defects in oocyte maturation, altered fertilization, disrupted meiosis, and mitochondrial dynamics abnormalities, leading to abnormal embryo preimplantation and lower birth rates [8,10,14,15,16]. There is a great deal of evidence that PCOS women with central obesity have higher levels of IR, dyslipidemia, and hyperandrogenism [7,17]. The temporary development of dyslipidemia and IR is a normal physiological response during pregnancy. Obese/overweight PCOS patients may face an additional risk of developing gestational complications related to these metabolic disorders [10]. For example, dyslipidemia has a negative impact on IVF clinical outcomes. A high pre-pregnancy weight is associated with an increased risk of pregnancy-induced hypertension, gestational diabetes, urinary infection, macrosomia, increased hospitalization, and Caesarean section and thus increased complexity of surgery (prolonged operative time and an increased risk of bleeding, wound infection, and wound dehiscence), increased difficulty of loco-regional anesthesia, and the increased use of general anesthesia, resulting in increased intraoperative mortality rates.

A maternal condition of obesity/overweight has, in addition, detrimental consequences for the life of the unborn child (obesity, coronary heart disease, stroke, diabetes mellitus, asthma, cognitive and attention dysfunction, mood and sleep disorders, and aggression). This may be due to an “obesiogenic” intrauterine environment and an “obesiogenic” home environment. Improvements in fertility (ovulation rates and menstrual regularity) and a higher rate of live births have been linked to weight loss, before undergoing IVF, among PCOS patients [10,18,19,20].

Thus, a BMI > 24.9 kg/m^2^ (especially when associated with PCOS) significantly reduces IVF success rates, increases risks during hormonal stimulation for IVF (OHSS, incongruous response to stimulation, and increased thrombotic risk), and increases the risk of obstetric and neonatal complications (abortion, malformative disorders, preterm delivery, placental abruption, intrauterine death, the premature rupture of membranes, fetal macrosomia, increased risk of operative vaginal delivery and caesarean section, and neonatal hypoglycemia) [1,21,22]. In addition to the aforementioned factors, obesity alone is a risk factor in pregnancy, predisposing individuals to arterial hypertension, pulmonary hypertension, left ventricular dysfunction, hypoxia, hypercapnia, hypercoagulability, the regurgitation and pulmonary aspiration of gastric contents, and hepatic steatosis [10,22]. Specifically, obese/overweight PCOS patients have an all-or-none response to hormonal stimulation aimed at IVF: hyperstimulation or a nonresponse. Moreover, such patients show low fertilization rates, poor embryo quality, and low pregnancy rates [8,15,23]. Weight loss is an effective treatment for PCOS; in fact, losing 5% or more of body weight reduces its symptoms and associated risks [9]. Several therapeutic strategies aim to reduce weight and body mass index (BMI), simultaneously improving reproductive functionality. Weight loss has been shown to improve reproductive outcomes by enhancing fertility, regularizing menstrual cycles, and increasing the likelihood of spontaneous ovulation/conception in anovulatory overweight/obese women [16]. This is probably due to the effects of a reduction in fat and/or lean mass, changes in some endocrinological and metabolic parameters, and a decrease in inflammatory status [8,24]. PCOS patients show increased inflammatory markers, C-reactive protein (CRP), interleukin 18 (IL-18), tumor necrosis factor α (TNF-α), interleukin 6 (IL-6), white blood cell count (WBC), monocyte chemoattractant protein-1 (MCP-1), and macrophage inflammatory protein-1α (MIP-1α). In fact, the adipose tissue (especially visceral) is considered an endocrine organ, producing and releasing several adipokines and cytokines (leptin, adiponectin, resistin, visfatin, TNF-α, and IL-6) [25]. These proinflammatory molecules result in the development of metabolic diseases. A condition of obesity/overweight determines endocrine mechanisms that alter inflammatory responses, interfering with both ovarian and endometrial function. Thus, the loss of fat mass (which is an endocrine organ) could cause a decrease in inflammatory status (by decreasing IL-6 and thus reducing CRP synthesis), resulting in improved reproductive function [24,26]. Pharmacological strategies are insulin-sensitizing drugs (metformin or D-chiro-inositol) and anti-androgens [7]. Metformin is efficient but has gastroenteric side effects and is considered a US Food and Drug Administration (FDA) pregnancy category B drug [7,27]. Lifestyle modifications (dietary intervention and exercise) should be the first-line treatment in overweight/obese PCOS patients because of the low potential costs, absence of adverse effects, and potential for long-term management [21,28]. There is no or limited evidence that one specific dietary composition is better than another in PCOS, but it seems that only low-carbohydrate diets can lead to a significant decrease in IR and in circulating markers of inflammation in overweight/obese PCOS patients [4,29]. The aim of this study was to assess the effects of different diet regimens, namely the very-low-calorie ketogenic diet (VLCKD) and Mediterranean diet (MD), on PCOS patients prior to IVF cycles, specifically regarding anthropometric, metabolic, and hormonal parameters and the safety of IVF techniques (rate of OHSS).

## 2. Materials and Methods

### 2.1. Study Design

The study was a nonrandomized controlled study in which 84 obese/overweight PCOS patients were recruited to evaluate a possible effect of two dietary therapies, VLKCD and Mediterranean, which they underwent before access to IVF techniques. Patients were evaluated at baseline and at 90 and 120 days since the beginning of both diets. The outcome measures were anthropometric measures (body mass index (BMI), waist and hip circumferences, waist-to-hip ratio (WHR), and abdominal circumference); hormone levels (luteinizing hormone (LH), follicle-stimulating hormone (FSH), FSH/LH ratio, estradiol, anti-Müllerian hormone (AMH), 17-alpha-hydroxy-progesterone, androstenedione, testosterone); metabolic parameters (cholesterol, high-density lipoprotein (HDL), triglycerides); Homeostasis Model Assessment (HOMA index); and reproductive parameters (menstrual regularity, antral follicle count (AFC), AMH, and risk of developing OHSS).

For FSH, LH, estradiol, 17-alpha-hydroxy-progesterone, androstenedione, and testosterone measurements, chemiluminescence immunoassay (CLIA) was performed, while AMH was analyzed by electrochemiluminescence immunoassay (ECLIA). A blood sample was taken within 5 days of the start of the menstrual cycle (early follicular phase). In patients with amenorrhea or oligomenorrhea, sampling was done randomly, or rather on any day of the menstrual cycle. During the evaluation of the hormone profile, the patients had not undergone IVF-related treatment with exogenous hormones.

Oligomenorrhea is a condition involving a menstrual interval of 35 days or more, secondary amenorrhea comprises a menstrual interval of 90 days or more, and polymenorrhea comprises a menstrual interval of less than 25 days [30]. The menstrual regularity of patients in an IVF center can be objectively assessed by transvaginal ultrasonography (echography ultrasound WS80A, Samsung; ultrasound probe V5-9, Samsung) or the study of the ovaries and subsequent endometrial changes. Notably, the objectivity of this examination was enhanced by the fact that it was performed by the same surgeon, a specialist in gynecology and obstetrics experienced in IVF.

AFC was assessed by transvaginal ultrasound (echography ultrasound WS80A, Samsung; ultrasound probe V5-9, Samsung, Suwon, South Korea) within 5 days of the start of the menstrual cycle (early follicular phase) or randomly in patients with amenorrhea. In the case of the patients recruited for the study, the ultrasound examination was always performed by the same surgeon previously mentioned, to avoid operator-dependent bias.

Body weight was measured to the nearest 0.1 kg using an electronic scale (INNOFIT INN-112) and height to the nearest 1 cm using a stadiometer (ADE MZ10042 Portable Stadiometer). BMI was calculated in kg/m^2^. All circumferences were assessed with a measuring tape. Waist circumference (WC) (cm) was measured in a standing position, keeping the abdomen relaxed, with the arms at the sides. The narrowest circumference was measured between the thighs and the iliac crest (generally a few centimeters above the navel). This is one of the indicators of abdominal body fat. The hip circumference (cm) was measured while standing. The operator stood sideways to take the maximum gluteal circumference. WHR was obtained by dividing the circumference of the waist by the circumference of the hips; this is another indicator of abdominal body fat. The thigh circumference (cm) was measured by having the subject stand, carrying their weight mainly on the limb that was not measured; the circumference was taken below the gluteal crease (higher circumference). The abdomen circumference was measured 2 cm below the navel. The patient was standing up and had to keep their abdomen relaxed. In order to limit measurement error, the measurements were performed three times, and the average value was considered.

### 2.2. Selection and Exclusion Criteria

In total, 84 patients seeking infertility treatment at the Complex Organizational Unit (UOC) of Reproductive Pathophysiology and Andrology, which carries out its activities within the Sandro Pertini Hospital (Rome, Italy), were recruited. Before starting the study (year 2022), all patients signed an informed consent form after being informed in detail about the study process. The study was approved by the local ethics committee “Lazio 2”, protocol 0137603/2022 (5 July 2022), and respected the Helsinki declaration. The patients participating in the study had the following characteristics: (i) age between 18 and 39 years; (ii) condition of overweight or obesity (BMI > 24.9 kg/m^2^) at the beginning of the treatment; (iii) diagnosis of polycystic ovary syndrome according to the criteria established in 2003 by the Rotterdam European Society of Human Reproduction and Embryology (ESHRE); (iv) nonuse of contraceptive methods during the experimental period; (v) motivation to lose weight; (vi) hyperinsulinemia.

During patient selection, exclusion criteria were based on all those concomitant diseases and conditions that could constitute a bias: pregnancy, lactation, liver, kidney and heart disease, episodes of gout or hyperuricemia, congenital adrenal hyperplasia, androgen-secreting tumors, Cushing’s syndrome, decompensated thyroidopathies, hyperprolactinemia, diabetes mellitus 1 and 2, endometriosis, and the use of ongoing insulin-sensitizing drugs. An additional criterion for exclusion from the study was a lack of motivation to follow the proposed diet plan.

In addition, although the VLCKD represents a possible therapeutic strategy for lipedema, it should be noted that the presence of this condition was absent in all patients recruited [31,32]. Since there are no known blood or urinary biomarkers or specific diagnostic tests for the diagnosis of lipedema, the assessment was performed through the patients’ history, visual inspection, and physical examination. The following criteria were evaluated by the nutritional biologist and gynecological surgeon: the excessive, bilateral, and symmetrical presence of subcutaneous adipose tissue between the iliac crest and malleoli and on the arms; the abrupt cessation of such enlargement of the lower and upper limbs at the wrists and ankles; the loss of the concave spaces on either side of the Achilles tendon; the presence of bruising; alterations in the appearance, temperature, and texture of the skin; abnormal gait and limited mobility; negative Stemmer’s sign; and punctate edema and/or chronic venous insufficiency [33,34].

The 84 patients were divided into two study groups: 42 patients received VLCKD therapy treatment and 42 patients received Mediterranean diet therapy treatment. After careful counseling, patients chose whether to follow the diet regimen or not and, if they did, they chose which one (VLCKD or Mediterranean). During counseling, the patient was informed that a BMI value > 24.9 kg/m^2^ (especially when associated with PCOS) significantly reduces IVF success rates, increases risks during hormonal stimulation aimed at IVF, and increases the risk of obstetric and neonatal complications. In addition, patients were thoroughly informed about all the diet’s features. However, under the internal regulation of the UOC, hormone stimulation aimed at IVF cannot be carried out in patients with BMI > 30 kg/m^2^. Only the female component of infertility was considered, while the male infertility factor was neglected. Therefore, embryo quality, implantation rates, and pregnancy outcomes were not evaluated.

All data were analyzed at baseline before the beginning of both diets and are summarized in Table 2. Statistical analysis (Student’s t test) found no statistically significant difference between the two groups. In this regard, the groups appeared to be homogeneous at baseline.

### 2.3. Ovarian Stimulation Protocol

For all patients, the protocol induced follicular growth by gonadotropin stimulation (mild stimulation with starting dose of 150 IU of recombinant gonadotropin), adding a GnRH antagonist at day 5 of stimulation. Each patient was monitored from day 5 of stimulation on alternate days by dosing estradiol and progesterone and via ultrasound monitoring for possible dosage adjustments during therapy. When the ultrasound and hormone parameters were found to be likely compatible with oocyte maturity (follicular diameter 18–20 mm and 200 pg/mL of E2 per follicle, presumably containing mature oocytes in MII stage), the trigger was set and oocyte pick-up was performed 32–36 h after trigger administration. The usual trigger drug is recombinant hCG (250 mcg). In case of OHSS risk, the drug administered is GnRH agonist (0.2 mg/mL). The route of administration of all the aforementioned drugs was subcutaneous and the patients could perform it independently.

### 2.4. Objectives

The objectives of the study were multiple: (i) the evaluation of the speed with which the adopted diet leads to the improvement of anthropometric parameters; (ii) the evaluation of the effects induced by different dietary therapies on metabolic and hormonal parameters affecting PCOS patient fitness and reproductive outcomes; and (iii) the evaluation of the incidence of the risk of developing ovarian hyperstimulation syndrome (OHSS) during hormone stimulation aimed at intracytoplasmic sperm injection (ICSI) in obese/overweight patients undergoing diet therapy.

### 2.5. Diet Therapies

The VLCKD group initially followed the “intensive phase” protocol for 60 days; during this period, patients used substitute foods, such as protein and carbohydrate sources already balanced specifically for this regimen, to which fresh non-starchy vegetables and some accompanying foods (roast chicken and turkey, bresaola, ham, salmon, tuna or swordfish) were added at each main meal. The substitute foods used for the VLCKD contained plant-based and animal proteins. Whey protein appears to substantially contribute to the preservation of muscle performance [35]. The sources of plant protein were peas and soybeans. Regarding the possible harmful effects of soy, all patients accessing the IVF center are screened (breast ultrasound, mammogram, and Papanicolaou test) to diagnose pre-cancerous or cancerous lesions in estrogen-sensitive tissues. This is because patients who access our center to undergo IVF will undergo hormone therapy (gonadotropins) with significantly increased levels of 17-beta estradiol. Our study patients, therefore, had no contraindications to soy intake. Vegetables were divided into unlimited and limited consumption, as described in Appendix A. During the first 3 days, we added medium-chain triglyceride (MCT) oil, which promotes and accelerates entry into ketosis under conditions of glucose restriction and consequently limits the effects of keto-adaptation [36]. The use of MCT oil has been discontinued because ketosis must be generated entirely by the oxidation of the endogenous lipids (excess adipose tissue). MCT oil was optional according to the clinician’s opinion and had the function of speeding up the patients’ entry into ketosis. Evaluation of the entry into ketosis was assessed by test strips for the measurement of ketone bodies in urine. The consumption of fruit juices, sugary drinks, and alcohol was forbidden. A maximum of 2 tablespoons of EVO per day was allowed. The use of vinegar, pepper, lemon, and spices was possible. The consumption of at least 2 L of water per day was strongly recommended. Other supplements, explained in Appendix A, were optional according to the clinician’s opinion and doses were administered via 2 daily sachets. Supplements were prescribed to all patients, under the recommendation of the nutritional biologist and the gynecological surgeon, because the VLCKD’s limited intake of fruits and vegetables does not cover the daily requirement of vitamins and minerals. In addition, the intake of magnesium and potassium helped to avoid muscle cramps, which are one of the possible side effects of the VLCKD. The daily low-calorie intake was 800 kcal; nutrient intakes were 20% carbohydrates (25 g), 50% protein (67 g), and 30% lipids (50 g). The nutrition plan was divided into 4 meals, schematized in Appendix A: breakfast (20% kcal daily), lunch (35% kcal daily), snack (10% kcal daily), and dinner (35% kcal daily). Appendix A show the nutritional information panels of the substitute foods.

To preserve the results obtained, the VLCKD’s intensive phase was followed by a 30-day transition period in which there was an increase in daily kcal in the VLCKD group. The daily calorie intake was 800 kcal from Monday to Friday, 1300 kcal on Saturday, and 1400 on Sunday. A maximum of 2 tablespoons of EVO per day was allowed. The use of vinegar, pepper, lemon, and spices was possible. The consumption of at least 2 L of water per day was strongly recommended. The type of diet in this phase was a hypocaloric Mediterranean regimen with 4 meals per day: breakfast, lunch, snack, and dinner (schematized in Appendix A). The Mediterranean group consisted of patients who preferred to adhere to traditional nutritional education. The diet therapy was a low-calorie Mediterranean regimen, in which the daily calorie intake was 1400 kcal for 90 days. The composition was 55% carbohydrates (10% simple ones), 15% proteins, and 30% lipids. The nutrition plan was divided into 5 meals (schematized in Appendix A): breakfast (20% kcal daily), snack (10% kcal daily), lunch (30% kcal daily), snack (10% kcal daily), and dinner (30% kcal daily). Rice and pasta were allowed twice a week. The use of vinegar, pepper, lemon, and spices was possible. The consumption of at least 2 L of water per day was strongly recommended. The higher calorie intake in this diet is due to the lack of the appetite suppressant effect that can be achieved with the VLCKD regimen.

### 2.6. Data Collection and Data Analysis

Parameters were collected by the staff of the UOC of Reproductive Pathophysiology and Andrology. The UOC deals with infertile couples from the diagnostic stage to the therapeutic indication and to the performance of the IVF technique. Diagnosis of this syndrome occurs during preliminary meetings, which then lead to the decision to possibly perform an IVF technique. In the case of overweight/obese PCOS patients, during the first medical examination, a routine assessment of anthropometric indices and metabolic parameters is performed, because eventual alterations in them could be associated with the syndrome. During counseling, the patient is made aware of the worsening effects of overweight/obesity on PCOS itself, on the success of IVF, and on the complications that can occur during hormonal stimulation aimed at IVF and on pregnancy (adverse effects on the pregnant woman, the fetus, and later on the newborn) [1,21,22]. Therefore, the patient may decide to embark on a nutritional course aimed at BMI normalization. Data related to the IVF cycle (hormonal stimulation, follicle development, hormone dosages of estradiol and progesterone during stimulation, the development of an OHSS risk condition, the number and quality of retrieved oocytes) are recorded in the couple’s medical record.

From this database, the patients were recruited to the study. Recruitment took place after the positive response of the Ethical Committee of the Asl Roma 2 hospital company on 5 July 2022. At this point, the use of the data was authorized and the diet therapies were proposed to patients.

Statistical analyses of the obtained parameters were performed with the Statistics Kingdom (2017) software and GraphPad Prism 9.2.0. Ordinal variables are presented as mean ± standard deviation (SD), while categorical variables are described as absolute frequencies. The normal distribution of data was checked with the Shapiro–Wilk test. For comparison between two groups, we performed the *t* test for normal data and the Mann–Whitney test for non-normal data. The χ^2^ test with a *p*-value of 0.05 was used for parameters with categorical variables. The statistical significance threshold was set to 0.05.

## 3. Results

### 3.1. Anthropometric Data

Anthropometric measurements, as shown in Figure 1, revealed a greater BMI loss (13.16 ± 7.37 kg/m^2^ in 90 days, 17.03 ± 8.79 kg/m^2^ in 120 days) in PCOS patients undergoing VLCKD than in PCOS patients undergoing the Mediterranean diet (7.29 ± 5.41 kg/m^2^ in 90 days, 8.94 ± 6.28 kg/m^2^ in 120 days) (*p* value < 0.0001 in 90 and 120 days, respectively, *t* test and Mann–Whitney test). The difference in BMI loss, according to the two treatments, can also be found in particular anthropometric indices, such as waist (9.20 ± 6.31 cm in 90 days and 12.27 ± 8.13 cm in 120 days for VLCKD; 4.14 ± 3.82 cm in 90 days and 5.10 ± 4.71 cm in 120 days for Mediterranean; *p* value < 0.0001 for both, Mann–Whitney test), hip (8.85 ± 6.81 cm in 90 days and 10.90 ± 8.20 cm in 120 days for VLCKD; 4.65 ± 3.63 cm in 90 days and 5.87 ± 4.79 cm in 120 days for Mediterranean; *p* value < 0.0001 in 90 days and *p* value < 0.001 in 120 days, Mann–Whitney test), and abdomen circumferences (8.23 ± 6.64 cm in 90 days and 11.05 ± 8.80 cm in 120 days for VLCKD; 4.31 ± 3.38 cm in 90 days and 5.48 ± 4.02 cm in 120 days for Mediterranean; *p* value < 0.001 in 90 days and *p* value < 0.05 in 120 days, Mann–Whitney test), and WHR (0.02 ± 0.06 in 90 days and -0.03 ± 0.07 in 120 days for VLCKD; 0.00 ± 0.04 in 90 days and 0.00 ± 0.04 in 120 days for Mediterranean; *p* value < 0.05 in 120 days, Mann–Whitney test).

### 3.2. Metabolic Profile

The metabolic profile seemed to vary according to the diet therapy: after VLCKD therapy, as shown in Figure 2, cholesterol and triglyceride levels and the HOMA index decreased, while HDL levels increased. Cholesterol levels decreased more in VLCKD patients (−16.10 ± 21.02 mg/dL in 90 days; −25.81 ± 30.07 mg/dL in 120 days) than in Mediterranean diet patients (−9.43 ± 18.19 mg/dL in 90 days; −15.29 ± 21.62 mg/dL in 120 days; *p* value < 0.05 for both, Mann–Whitney test).

In contrast, triglyceride levels appeared to decrease more in the VLCKD group only at 90 days (−17.05 ± 24.23 mg/dL in VLCKD, −13.57 ± 36.30 mg/dL in Mediterranean; *p* value < 0.05, Mann–Whitney test), whereas, at 120 days, there appeared to be no statistically significant difference.

A greater increase in HDL levels was observed at 120 days in VLCKD patients (4.87 ± 8.59 mg/dL) than in Mediterranean diet patients (7.08 ± 8.79 mg/dL) (*p* value < 0.05, *t* test).

Moreover, the HOMA index decreased more in VLCKD patients (−1.45 ± 1.31 in 90 days; −1.47 ± 2.41 in 120 days) than in Mediterranean diet patients (−0.33 ± 1.60 in 90 days; 0.26 ± 3.08 in 120 days; *p* value < 0.001 at 90 days and *p* value < 0.05 at 120 days, *t* test for both).

### 3.3. Hormone Profile

Regarding the hormone profile, no statistically significant differences were observed, after 90 and 120 days of treatment, in serum variations in the concentration of FSH (1.04 ± 2.03 mUI/mL in 90 days and 1.60 ± 2.70 mUI/mL in 120 days for VLCKD; 0.37 ± 1.81 in 90 days and 0.65 ± 2.00 mUI/mL in 120 days for Mediterranean; *p* value > 0.05, *t* test) and LH (−0.84 ± 3.72 mUI/mL in 90 days and −1.40 ± 5.12 mUI/mL in 90 days for VLCKD; −0.94 ± 2.87 mUI/mL in 90 days and −0.81 ± 3.24 mUI/mL in 120 days for Mediterranean; *p* value > 0.05, Mann–Whitney test).

However, the FSH/LH ratio showed a statistically significant difference as a function of treatment at 120 days (0.47 ± 0.96 in VLCKD patients, 0.15 ± 0.48 in Mediterranean patients; *p* value < 0.05, Mann–Whitney test), but not at 90 (0.20 ± 0.56 in VLCKD patients, 0.11 ± 0.46 in Mediterranean patients; *p* value > 0.05, *t* test). The same trend was exhibited by the levels of 17-alpha-hydroxy-progesterone (−0.20 ± 0.64 ng/mL in 90 days and −0.36 ± 0.75 ng/mL in 120 days for VLCKD; −0.01 ± 0.34 ng/mL in 90 days and −0.07 ± 0.35 ng/mL in 120 days for Mediterranean; *p* value < 0.05 in 120 days, Mann–Whitney test).

In contrast, AMH levels appeared to change as a function of the diet therapy at 90 days (−1.79 ± 2.05 ng/mL for VLCKD and −0.92 ± 1.61 ng/mL for Mediterranean; *p* value < 0.05, *t* test) but not at 120 (−2.23 ± 2.28 ng/mL for VLCKD and −1.38 ± 2.05 ng/mL for Mediterranean; *p* value < 0.05, *t* test).

Estradiol levels, on the other hand, did not appear to alter according to the diet therapy followed by the patients at either 90 (1.08 ± 21.78 pg/mL in VLCKD, −0.45 ± 25.25 pg/mL in Mediterranean; *p* value > 0.05, *t* test) or 120 days (8.06 ± 28.08 pg/mL in VLCKD, 5.22 ± 32.89 pg/mL in Mediterranean; *p* value > 0.05, *t* test).

Interesting results, shown in Figure 3, were obtained at both 90 and 120 days of diet therapy in terms of androstenedione (−0.54 ± 0.47 ng/mL in 90 days and −0.73 ± 0.60 ng/mL in 120 days for VLCKD; −0.27 ± 0.50 ng/mL in 90 days and −0.19 ± 0.87 ng/mL in 120 days for Mediterranean; *p* value < 0.5, *t* test for both) and testosterone levels (−0.40 ± 0.49 nmol/mL in 90 days and −0.62 ± 0.78 nmol/mL in 120 days for VLCKD; −0.03 ± 0.36 nmol/mL in 90 days and −0.12 ± 0.48 nmol/mL in 120 days for Mediterranean; *p* value < 0.001, *t* test at 90 days, *p* value < 0.05, Mann–Whitney test) and antral follicle count (AFC) (−4.31 ± 3.23 in 90 days and −5.98 ± 4.16 in 120 days for VLCKD, −1.95 ± 2.45 in 90 days and −2.95 ± 3.10 in 120 days for Mediterranean; *p* value < 0.001, *t* test for both times).

### 3.4. Menstrual Regularity

Considering the change in menstrual regularity condition in PCOS patients after 90 and 120 days of the VLCKD and Mediterranean diet, a statistically significant difference was observed regarding the percentages of improvement from the initial condition in 90 (*p* value < 0.05, χ^2^ test) and 120 days (*p* value < 0.001, χ^2^ test).

As shown in Figure 4A, VLCKD treatment resulted in an improvement from the initial condition in 50% and 71% of cases, respectively, in 90 and 120 days; there was no change from the initial condition in 50% and 29% of cases, and a worsening condition was not observed in any of the patients under study. In contrast, Mediterranean diet therapy resulted in an improvement from the initial situation in 17% and 26% of cases, respectively, in 90 and 120 days; in no change in 81% and 71%; and in a worsening condition in 2% of cases.

In addition, both the success and failure of the treatment concerning the menstrual parameter were evaluated. As success was considered for any patient who, after 90 and 120 days from the start of treatment, presented a condition of regularity or irregularity (but starting with the amenorrhea condition); as failure, we considered a condition of amenorrhea or irregularity (starting from a pre-existing condition of irregularity). Again, a statistically significant difference was observed in the effect on the success and failure rates obtained following the two diet therapy treatments at both times (*p* value < 0.05 in 90 days, *p* value < 0.0001 in 120 days, χ^2^ test). As shown in Figure 4B, the ketogenic diet treatment resulted in a success rate of 74% in 90 days and 95% in 120 days, while the Mediterranean diet treatment resulted in a success rate of 48% in 90 days and 59% in 120 days.

### 3.5. Risk from Ovarian Hyperstimulation Syndrome and Thromboembolic Complications

An analysis of the risk of developing OHSS in PCOS patients according to treatment revealed a statistically significant difference: as shown in Figure 5, patients who followed the VLCKD had a 27% risk rate of OHSS, while patients who followed the Mediterranean diet had a 68% risk rate of OHSS (*p* value < 0.05, χ^2^ test).

None of the patients developed thromboembolic complications during stimulation.

## 4. Discussion

Overweight/obesity and PCOS influence each other, probably also due to an increase in obesity-related inflammatory factors [16]. We evaluated the changes in anthropometric parameters as a function of different diet therapies, to determine the implications for the patient’s reproductive health. In the context of IVF, the time factor is essential, because female fertility progressively decreases as age increases. Therefore, the time range in which the results are obtained is a key element. Our data indicate that both the Mediterranean diet and VLCKD showed interesting results regarding an improvement in anthropometric parameters.

Specifically, the VLCKD in PCOS patients appeared to lead to better anthropometric outcomes in a reduced time, particularly in those areas where central adiposity was concerned. As we hoped, these improvements were mainly concentrated in the BMI, WHR, and abdomen, waist, and hip circumferences. The BMI, while not a direct measure of adiposity, does provide a practical means to estimate body fat mass. Anthropometric indices, such as WC and WHR, estimate the body composition and, in particular, the visceral fat [37]. Thus, BMI, WC, and WHR represent a sufficiently reliable estimate of the body fat distribution that is non-invasive, low-cost, repeatable, and easy to measure. The gold standards for the quantification of visceral fat are magnetic resonance, computed tomography, dual-energy X-ray absorptiometry (DXA), and bioimpedance, but the equipment is not available in every hospital facility [38,39,40,41,42]. The results of the VLCKD can be attributed to the reduced calorie intake, which is possible because of the satiating effect of protein and the appetite suppression induced by ketosis [41,43]. Several pieces of evidence show that the VLCKD is characterized by a protective effect toward muscle mass, and different mechanisms have been proposed to support this theory: increased adrenergic stimulation, the braking effect of ketone bodies on muscle protein breakdown, and the stimulation of muscle protein synthesis due to increased growth hormone (GH) and protein intake from the diet. Despite this, the study, focusing only on the assessment of anthropometric parameters, does not allow us to exclude that the increased loss of mass is caused by the loss of muscle mass and/or fat mass [44,45,46]. However, it must be emphasized that all anthropometric parameters had a non-negligible standard deviation. This could be due to the differences in the adherence of patients to the dietary regimens, but it represents a major limitation of this study. The future implementation of a sample unit could resolve this issue. The results obtained with the VLCKD, and, to a lesser extent, the Mediterranean diet, suggest that these nutritional approaches lead to weight reduction and to an improvement in the condition of central adiposity within a very narrow time range. Obesity represents a state of chronic inflammation, which can lead to the development of insulin resistance [47]. The VLCKD allows patients to tolerate a very low amount of kcal. However, the difference in this dietary pattern, which makes it more suitable for women suffering from PCOS, is in the distribution of nutrients, with a greater intake of proteins compared to carbohydrates.

A loss of central metabolically active fat could therefore lead to a break in the typical vicious cycle of PCOS. Our results show that greater weight loss, and thus greater visceral fat loss, is accompanied by a significant improvement in the metabolic profile in VLCKD PCOS patients. The improvement in the metabolic profile (increase in HDL and decrease in triglycerides, cholesterol, and HOMA index) achieved following the VLCKD could lead to an interruption in the vicious cycle of pathological events associated with PCOS, or at least to its reduction. The latter seems to be effectively reflected in the marked improvement in the reproductive health of PCOS patients during IVF preparation.

In fact, the VLCKD seems to be related to an improvement in reproductive health (menstrual regularity, lower AFC, and reduced risk of developing OHSS) in PCOS patients, due to the modification of key metabolic pathways, as reported in a recent study [48]. Restoring menstrual regularity is very important for the less complicated management of the patient’s ovulation and, at the same time, allows the possibility of intervention through targeted intercourse. Meanwhile, decreasing AFC and the risk of developing OHSS allows the easier and safer management of the patient during hormonal stimulation, aimed at IVF. The VLCKD-induced reduction in the risk of OHSS is likely due to the lowered antral follicle count, which, together with PCOS, is a risk factor for the development of the syndrome. Furthermore, we draw attention to the fact that none of the patients developed thromboembolic complications during stimulation. This was linked, in addition to the reduction in body weight, to the careful screening to which we subject all our patients before starting therapy aimed at IVF.

The increased safety profile could be related to the improved hormonal sexual profile following the VLCKD. Abdominal obesity in PCOS patients seems to be co-responsible for the development of hyperandrogenism due to the insulin-mediated overstimulation of ovarian steroidogenesis [7]. In our study, the increased weight loss in VLCKD PCOS patients was indeed accompanied by an increased FSH/LH ratio and decreased androstenedione and testosterone levels. Regarding the AMH value, our data showed a reduction in the value with the VLKCD diet, but these data are in contrast with a previous work [49]. This difference might be caused by the distinct populations examined; in particular, our group was only composed of PCOS patients, who usually present higher values of AMH, with respect to the obese patient group in the other work, which was not investigated regarding the PCOS condition.

Moreover, our results are in accordance with a recent study by Magagnini et al., in which VLKCD patients showed a significant reduction in BMI, WC, the HOMA index, and serum AMH levels, as was the case in our cohort of VLKCD patients [50]. These results might be explained also by the positive impact of the polyunsaturated fatty acid composition presented in both diets. Such fatty acids can stimulate carbohydrate metabolism and insulin sensitivity, improving the BMI, HOMA index, and metabolic–hormonal profile, leading to lower lipogenesis and a reduced inflammation status [51]. Despite this fact, we did not analyze specifically the contribution of fatty acids during weight reduction; the amount of fatty acids included are presented in additional Appendix A.

Our data confirm the VLCKD diet as an optimal choice for preparation for entry into IVF, especially because of the time range in which these results are achieved and in relation to the patient’s safety and health. The results could be due to the loss of lean and/or fat mass, changes in some endocrinological and metabolic parameters, and/or a decrease in inflammatory status [8,24]. Indeed, the VLCKD group showed greater weight and BMI loss, particularly in those body areas where central adiposity is concentrated. Due to the low glycemic load, the VLCKD may improve hyperglycemia, insulin sensitivity, dyslipidemia, and blood pressure, resulting in a reduction in inflammatory status [52]. In fact, lower carbohydrate intake, which is possible in the VLCKD diet but not in the Mediterranean diet, results in reduced inflammation and oxidative stress. Simple carbohydrates result in the de novo synthesis of simple fatty acids, leading to both lipotoxicity and hyperinsulinemia. This results in the nuclear translocation of NF-kB, the extracellular release of proinflammatory mediators from macrophages, and, finally, systemic insulin resistance, resulting in systemic inflammation. Indeed, in a high-fat diet, as demonstrated in a mouse model, obesity triggers inflammation in the adipose tissue, leading to organ damage, such as hepatic inflammation [53].

Thus, the relationship between the inflammatory state and IR is caused by the action of proinflammatory cytokines released by macrophages in the adipose tissue, which block the action of insulin in various tissues, including the adipose tissue, liver, and skeletal muscle. The enhancement in hyperinsulinemia could have several favorable effects on reproductive function because (i) it has both direct and indirect effects on the folliculogenesis and steroidogenesis of granulosa and theca cells driven by intraovarian gonadotropins and (ii) it results in a more favorable biochemical environment in the ovaries [24,25]. Thus, weight/BMI loss induces both a decrease in the inflammatory state and an improvement in IR. This could regulate both hormonal imbalance and ovarian function. Thus, addressing the obesity/overweight condition could lead to a decrease in the risk of abortion, gestational diabetes mellitus, pregnancy-induced hypertension, preeclampsia, and poor infant outcomes [1,6,7]. Therefore, VLCKD diet therapy in obese/overweight PCOS patients could decrease both infertility problems and gestational and/or neonatal complications.

In addition, the drastic body change has a noticeable effect on the patient’s degree of satisfaction. Indeed, all patients in both groups adhered to the diet regimens for the whole study period of 120 days and none of them reported side effects or complaints, and this may favor the maintenance of an appropriate lifestyle.

## 5. Conclusions

The importance of pre-pregnancy nutritional strategies must be evaluated considering the worldwide trend of increasing obesity, which today can be considered a global epidemic [8,10,17]. This trend could lead to an increase in both the incidence and severity of PCOS. Considering that the onset of the syndrome occurs in adolescence and the increased incidence of obesity is mainly found in childhood, these issues need to be evaluated to avoid the exacerbation of the symptomatology typical of PCOS. Fortunately, lifestyle changes improve metabolic abnormalities and symptomatology [2,3]. Hence, the Mediterranean diet could be an optimal nutritional education strategy, as well as the VLKCD, which, in a previous study, was shown to be more effective than a standard low-calorie diet [54]. Despite this, a “fertility diet” has not been yet identified. This is because each person has a unique genome, proteome, metabolome, microbiome, and exposome, and the aim of future studies and nutritional strategies will be toward personalized medicine and nutritional support, tailored to individual features. However, we chose these two dietary regimens because they represent key strategies in obese/overweight patients, especially those suffering from PCOS, who have higher reproductive risks, as well cardiovascular and metabolic conditions, such as diabetes. The focus of the present study was not to emphasize one diet with respect to another, but to underline how different nutritional approaches can increase the likelihood of restoring fertility and lead to the success of IVF in obese/overweight PCOS patients. Indeed, weight reduction leads to better reproductive performance and is functional for a healthy pregnancy in such patients [8].

The results obtained with the different nutritional therapeutic approaches, particularly the improvements after only 120 days of treatment, are very interesting. The purpose of this study was to achieve weight loss in the shortest possible period, prior to IVF in patients affected by PCOS. Patients who attend an IVF center aim to achieve pregnancy, in most cases after years of unsuccessful attempts. Indeed, the proposal of long-term diet therapy could result in a high drop-out rate. In addition, for reproductive purposes, the patient’s age is crucial, and a greater patient age is counterproductive to the IVF program. Therefore, a dietary regimen that achieves a BMI compatible with access to IVF therapy in the short term (at our center, BMI < 30 kg/m^2^) results in greater patient compliance and is more suitable for the needs of an IVF center. As a result, the advantage of the VLCKD has been demonstrated from the perspective of the short time range in which the results are obtained; as reported by the American Society for Reproductive Medicine recently, the health benefits of postponing pregnancy (and IVF) to achieve weight loss must be balanced against the risk of declining fertility with advancing age. In addition, the VLCKD’s side effects (headache, muscle cramps, asthenia, and halitosis) were transient, moderate, and well tolerated [55].

An important outcome, concerning women’s health during IVF treatment, is the significant reduction in the risk of developing OHSS in VLCKD patients, compared with Mediterranean diet patients. This result could be related to the reduction in AFC and to the greater weight loss in such patients. However, the link between obesity/overweight and an increased risk of developing OHSS is still controversial [56,57]. Future works will be needed to clarify the molecular mechanisms modified by diet interventions that lead to improved symptomatology and reproductive success. In addition, weight loss positively impacts thrombotic risk up to the entire first trimester of pregnancy, as well as the risk of obstetric, neonatal, and postnatal complications and metabolic diseases [58,59,60,61,62].

Our data confirm that weight and BMI reduction results in improved fertility. Although the decision to postpone IVF treatment to allow weight loss often results in increased maternal age in patients who are no longer very young, a dietary approach, such as the VLCKD, allows them to achieve a better outcome in the shortest possible time.

## Figures and Tables

**Figure 1 nutrients-15-04444-f001:**
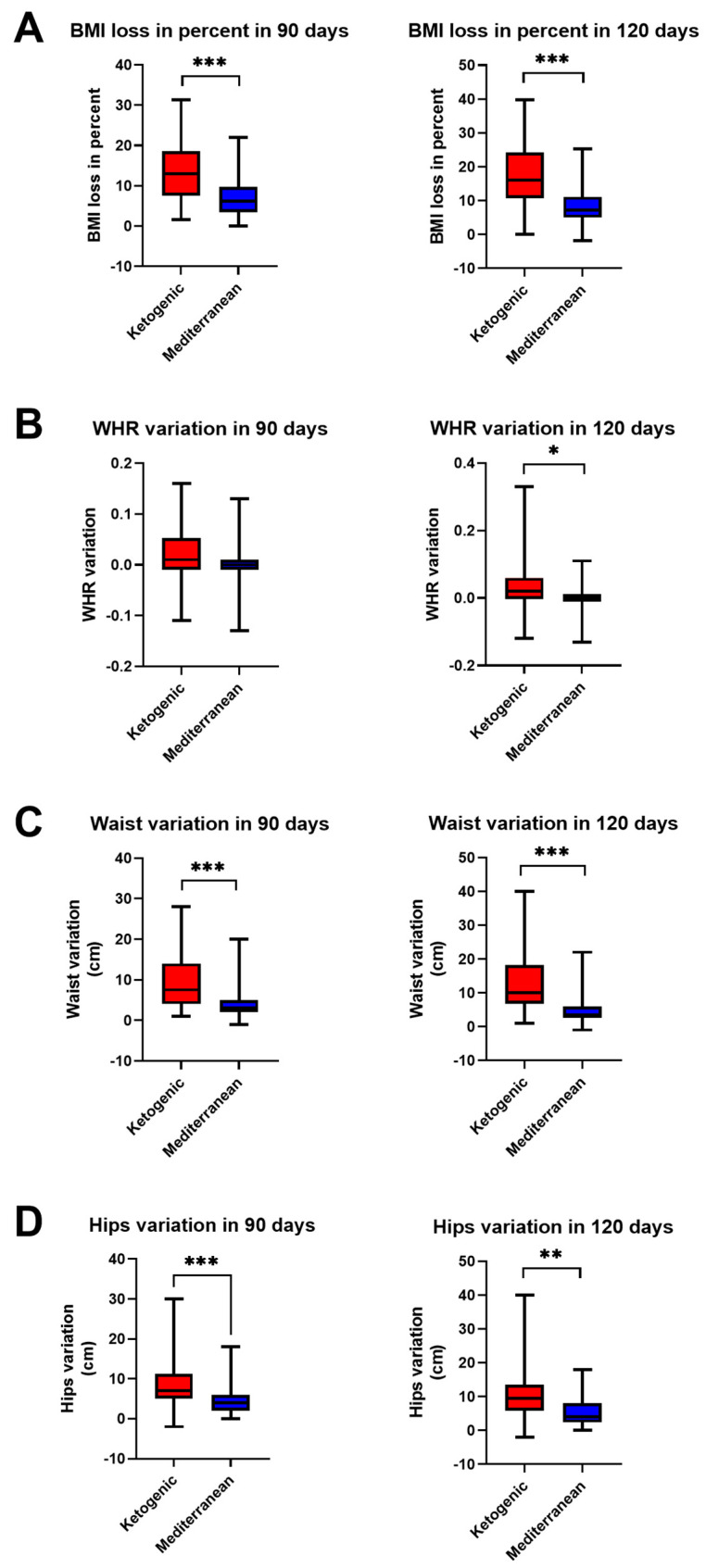
Change in anthropometric data in PCOS patients after 90 and 120 days of treatment with VLCKD (red) and Mediterranean (blue) diet. (**A**) BMI change, (**B**) WHR change, (**C**) waist circumference change, (**D**) hip circumference change. The standard deviation is shown in the figure. *** *p* < 0.0001; ** *p* < 0.001; * *p* < 0.05.

**Figure 2 nutrients-15-04444-f002:**
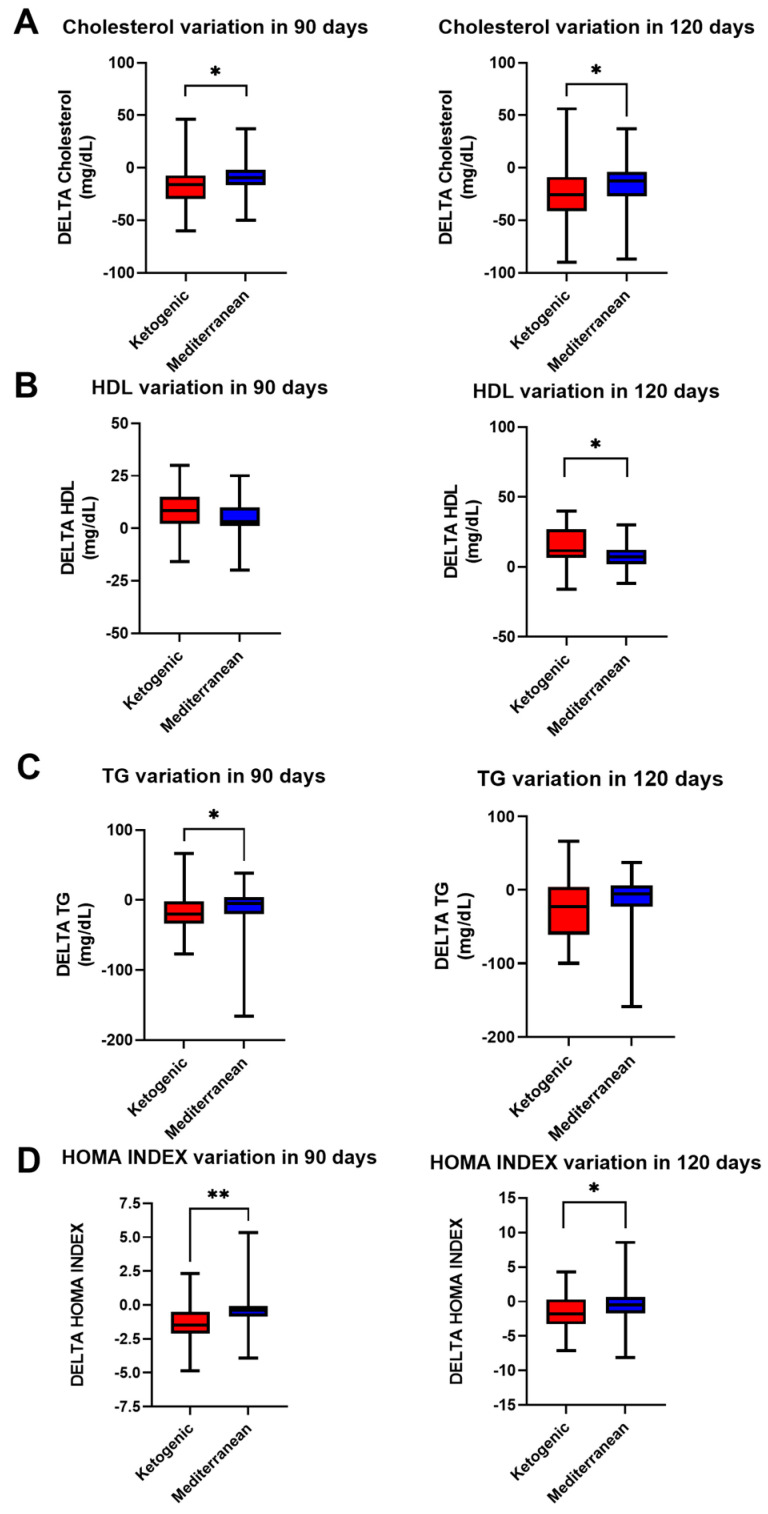
Change in metabolic profile in PCOS patients after 90 and 120 days of treatment with VLCKD (red) and Mediterranean (blue) diet. (**A**) Cholesterol variation, (**B**) HDL variation, (**C**) TG variation, (**D**) HOMA index variation. The standard deviation is shown in the figure. ** *p* < 0.001; * *p* < 0.05.

**Figure 3 nutrients-15-04444-f003:**
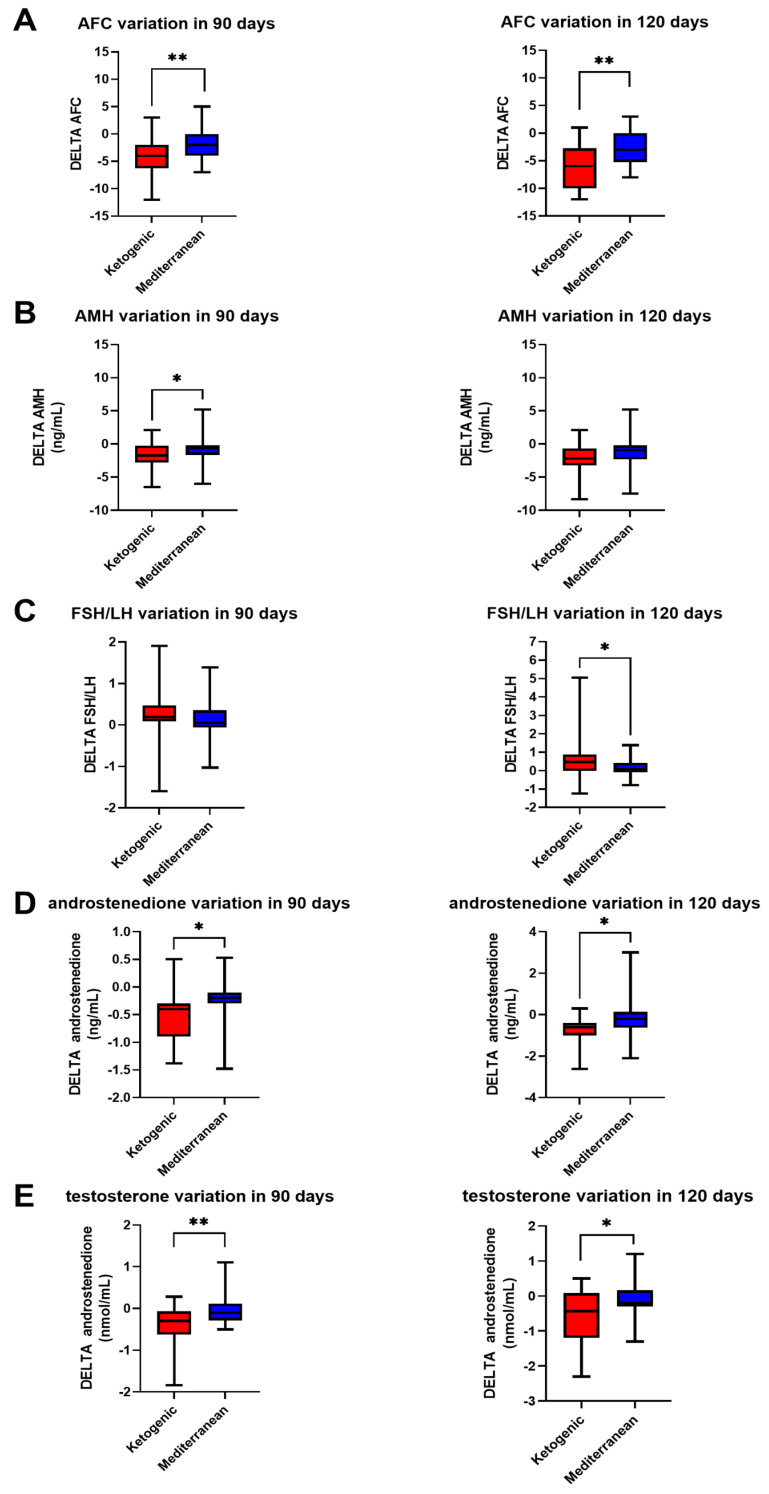
Change in sexual hormonal profile in PCOS patients after 90 and 120 days of treatment with VLCKD (red) and Mediterranean (blue) diet. (**A**) AFC variation, (**B**) AMH variation, (**C**) FSH/LH variation, (**D**) androstenedione variation, (**E**) testosterone variation. The standard deviation is shown in the figure. ** *p* < 0.001; * *p* < 0.05.

**Figure 4 nutrients-15-04444-f004:**
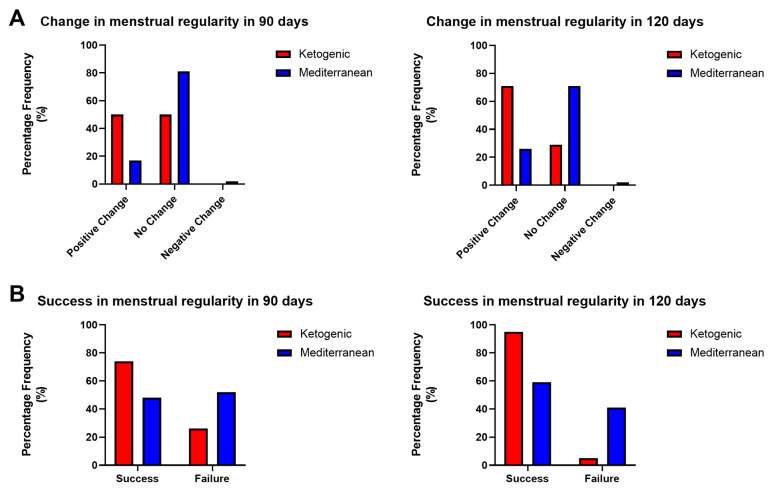
(**A**) Relative frequency of percentages of improvement, no change, and worsening in 90 and 120 days of treatment with VLCKD (red) and Mediterranean (blue) diet (*p* value < 0.05, χ^2^ test) and (**B**) percentages of success and failure in 90 and 120 days of treatment with VLCKD (blue) and Mediterranean (red) diet (*p* value < 0.05, χ^2^ test).

**Figure 5 nutrients-15-04444-f005:**
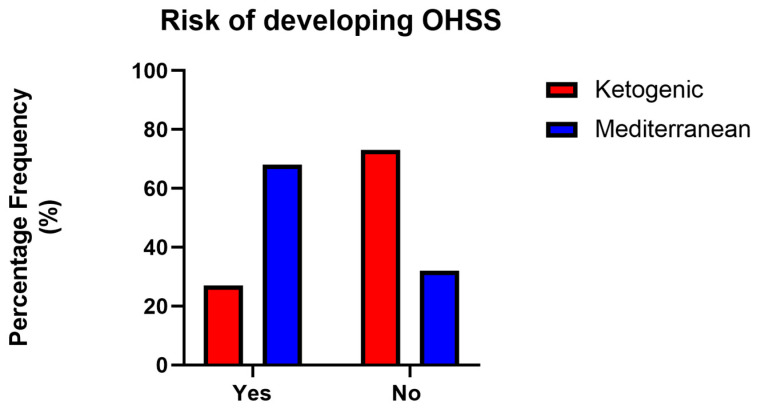
Assessment of the risk of developing OHSS following 120 days of VLCKD (red) or Mediterranean (blue) treatment.

**Table 1 nutrients-15-04444-t001:** BMI classification in women.

BMI (kg/m^2^)	Classification
<18.5	Underweight
18.5–24.9 25–29.9	Normal Weight Overweight
30–34.9	I Degree Obesity (Mild)
35–39.9	II Degree Obesity (Moderate)
>40	III Degree Obesity (Severe)

**Table 2 nutrients-15-04444-t002:** Patient parameters for VLKCD (42 women) and Mediterranean (42 women) groups, at time T 0. All data are expressed as mean and standard deviation (SD).

Demographic and Hormonal Parameters	VLCKD T 0	Mediterranean T 0
Age	33.88 ± 4.46	33.88 ± 3.61
BMI (kg/m^2^) Time of infertility (year)	31.23 ± 4.25 2.9 ± 1.2	28.65 ± 2.19 3.1 ± 1.7
WHR	0.88 ± 0.14	0.89 ± 0.08
FSH (mUI/mL)	6.55 ± 2.37	6.88 ± 2.70
LH (mUI/mL)	8.23 ± 4.28	7.90 ± 4.48
FSH/LH (mUI/mL) ratio	0.97 ± 0.53	1.10 ± 0.72
Estradiol (pg/mL)	51.09 ± 19.03	57.83 ± 42.90
AMH (ng/mL)	5.33 ± 3.09	6.08 ± 3.60
AFC	18.31 ± 4.85	18.33 ± 3.01
17-α-OH-progesterone (ng/mL)	1.07 ± 0.78	0.95 ± 0.64
Testosterone (mmol/L)	1.81 ± 1.35	1.76 ± 1.19
Androstenedione (ng/mL)	3.56 ± 0.64	3.15 ± 0.93
Cholesterol (mg/dL)	186.95 ± 36.40	195.50 ± 34.63
HDL (mg/dL)	47.14 ± 14.08	49.31 ± 11.91
Triglycerides (mg/dL)	121.43 ± 50.13	124 ± 54.91
HOMA index	2.27 ± 3.29	1.50 ± 1.98

## Data Availability

The data presented in this study are available on request from the corresponding author.

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
