# Peer review of "The Impact of Nutritional Therapy in the Management of Overweight/Obese PCOS Patient Candidates for IVF"

_nutrients, 2023, doi:10.3390/nu15204444_

Round 1

Reviewer 1 Report

Manuscript ID: nutrients-2640527
Title: The impact of nutritional therapy in management of overweight/obese
PCOS patients candidates to IVF
Authors: C Meneghini, C Bianco, F Galanti, V Tamburelli, A Dal Lago, E Licata, MGC Fabiani, R Corno, D Miriello, R Rago

These researchers evaluated the effects of Very-Low-Calorie Ketogenic diet (VLCKD) and Mediterranean Diet (MD) on body weight, hormonal and metabolic parameters in 84 obese/overweight PCOS patients. Results of this investigation suggest that VLCKD is an optimal choice which can improve anthropometric parameters as BMI, women’s reproductive health, restoring menstrual regularity and low risk of developing ovarian hyperstimulation syndrome (OHSS). Thus, VLCKD may help for entry into IVF.

The researchers extensively conducted several experiments including anthropometric parameters, metabolic and hormonal profiles. Furthermore, menstrual regularity was critically evaluated.

The manuscript is very well written and informative.

Author Response

thanks for your appreciation.

Best regards

Caterina Meneghini

Reviewer 2 Report

The effectiveness and safety of the in vitro fertilization procedure is an important issue that seems fully justified. Searching for procedures that facilitate good preparation for this method of infertility treatment is also of great application importance, especially for women at risk of failure and complications.

The study presented in the manuscript was well presented and described, but it would be worth adding information on the nutritional value, including nutrient content, of the diet actually consumed by the patients. What was dietary adherence like and did it vary during the 120 days of the program?

However, did it matter for the obtained effects that the energy deficit in women using VLCD was probably significantly greater than that in women on the Mediterranean Diet? The authors tried to explain these differences in the methodological part of the work, but it is also worth taking it into account in the discussion of the results. It is also worth referring to the characteristics of fatty acids in both types of diets and their possible impact on the determined parameters.

Could the results be influenced by the difference in the initial BMI value in both groups of women?

To confirm the obtained results, wouldn’t it be worth assessing the relationship between parameters related to female fertility and changes in anthropometric indicators during diet therapy?

A description of the statistical analysis for the presented parameters should be added to Table 2.

Additionally, the most important results could be presented in the summary. In the conclusions chapter (lines 551-553) it should probably be "standard low-calorie diet" instead of "standard low-carbohydrate diet".

Author Response

Thanks for your review and suggestions.

The study presented in the manuscript was well presented and described, but it would be worth adding information on the nutritional value, including nutrient content, of the diet actually consumed by the patients. What was dietary adherence like and did it vary during the 120 days of the program?

  • We have added information regarding gram’s quantity of carbohydrates, protein and lipid of VLKCD, instead the Kcal. % are already explained for both diets in line 290-313. Additional informations regarding nutritional content are rappesented in supplementary tables.
  • Regarding the dietary adherence during the 120 days of the program, in the last part of “conclusion” of the paper, we clarified that all patients had a full adherence to both diet regimes for the 120 days. Indeed, none of the women enrolled didn’t respect nutritional indications given or presented adverse effects due to diets.

However, did it matter for the obtained effects that the energy deficit in women using VLCD was probably significantly greater than that in women on the Mediterranean Diet? The authors tried to explain these differences in the methodological part of the work, but it is also worth taking it into account in the discussion of the results. It is also worth referring to the characteristics of fatty acids in both types of diets and their possible impact on the determined parameters.

-The VLCKD diet allows you to tolerate a very low amount of cKal. But the difference in this dietary pattern, which makes it more suitable for women suffering from PCOS, is the distribution of nutrients with a greater intake of proteins compared to carbohydrates.

  • We improved the discussion part of the results adding references 50-51 (substituting them with the two previous that were uncorrected). We clarified that fatty acids included in both types of diets, might have a positive impact on the determined parameters as weight reduction and metabolic features.

Could the results be influenced by the difference in the initial BMI value in both groups of women?

  • Although the initial BMI was slightly different in two groups, there was no statistically difference, as you can see in table II, and so on we think that our results are not influenced by this small difference.

To confirm the obtained results, wouldn’t it be worth assessing the relationship between parameters related to female fertility and changes in anthropometric indicators during diet therapy?

  • We added in the abstract the main results with statical significance.

  • The relationship between parameters related to female fertility and changes in anthropometric indicators during diet therapy are already explained in the discussion part, where we highlighted that in PCOS patients, diets inter lead to an improvement of metabolic profile, thanks to the breakdown of the vicious circle: the decrease of body weight lead to an improvement of the hormonal profile, and also of the AFC and AMH, and so on the menstrual regularity and ovarian cyclic function.

A description of the statistical analysis for the presented parameters should be added to Table 2.

  • We added a description of the statistical analysis for the presented parameters in Table 2. 

Additionally, the most important results could be presented in the summary. In the conclusions chapter (lines 551-553) it should probably be "standard low-calorie diet" instead of "standard low-carbohydrate diet".

  • Corrected as your suggestion.

Reviewer 3 Report

Manuscript has a good title. English language has good quality. figures have acceptable quality. There are some modifications that are essential to be exerted in the manuscript.

1. About page 4, line 147-151

Please mention the protocole by which sampling for assessment of FSH, LH, estradiol, AMH, 17-alpha-hydroxy-progesterone, androstenedione and testosterone were performed

+ Please explain that why sampling of patients with amenorrhea was done randomly? This was conducted based on which scientific reference? Is this method of sampling from patients with amenorrhea is validated? If yes, please explain about it.

2. About line 154-159 in page 4

This section needs proper citation.

3. About line 166 in page 4

Please explain why you measured BMI in this manuscript.

4. Line 449-450 in page 6

Please insert proper reference here

5. Please talk about related animal studies in section "Discussion"

6. Please explain that why you elected VLKCD and Mediterranean for your manuscript?

7. Please check and adjust the "Reference list" based on the regulations of reference list of journal. (Titles, doi, the name of journal and ... )

Author Response

Thanks for your comments and review.

 About page 4, line 147-151. Please mention the protocole by which sampling for assessment of FSH, LH, estradiol, AMH, 17-alpha-hydroxy-progesterone, androstenedione and testosterone were performed

We mentioned the laboratory protocols by which sampling for assessment of FSH, LH, estradiol, 17-alpha-hydroxy-progesterone, androstenedione and testosterone were performed, as well for AMH, and those is describe in line 147-149.

Please explain that why sampling of patients with amenorrhea was done randomly? This was conducted based on which scientific reference? Is this method of sampling from patients with amenorrhea is validated? If yes, please explain about it.

We are sorry for this misunderstanding, we mean that PCOS patients have usually might have amenorrhea, that is feature of PCOS according to Rotterdam Criteria, and so on the sampling was carried out in any day of the menstrual period, this means “randomly”. Furthermore, we don’t want to give external hormones or medical treatments, that can impact with the analyzed ones (FSH, LH). We explained this in the text too.

About line 154-159 in page 4. This section needs proper citation.

Regarding line 154-159 in page 4 we didn’t put a citation cause is only an explanation of two medical condition, characteristic of PCOS condition, which is already described in first lines of the paper.

About line 166 in page 4. Please explain why you measured BMI in this manuscript.

About line 166 in page 4 we measured BMI in this manuscript cause is a very important anthropometric parameter, very easy to measure and compare, and BMI plays a key role in the pathophysiology of PCOS. BMI is not the only anthropometric parameter considered. The circumferences of the waist, hips, abdomen and the waist/hip ratio are very important parameters because they allow us to evaluate the distribution of body fat. Visceral adipose tissue is, in fact, a metabolically active tissue that plays a fundamental role in the etiopathogenesis of PCOS.

 Line 449-450 in page 6. Please insert proper reference here

“Besides the hormone disorders and subfertility that are common in the polycystic ovary syndrome (PCOS), in obesity the adipocytes act as endocrine organ. The adipose tissue indeed, releases a number of bioactive molecules, namely adipokines, that variably interact with multiple molecular pathways of insulin resistance, inflammation, hypertension, cardiovascular risk, coagulation, and oocyte differentiation and maturation”. This is an extract from reference number 16 in the bibliography. We will insert the correct reference.

Please talk about related animal studies in section "Discussion"

We added a reference related animal studies in section "Discussion", with references 53.

Please explain that why you elected VLKCD and Mediterranean for your manuscript? 

Both diets represent key strategies in obese/overweight patients, especially those suffering PCOS, that have higher risk for the health as higher cardiovascular and metabolic disease as Diabetes. Unfortunately, in Italy and in industrial countries, also due to globalization, both men that women are losing the habit of using a healthy and balanced diet. The Mediterranean diet is safe, low cost and above all it is considered a world heritage. Among the other diets, the VLKCD diet also represents a safe diet if performed for excessively not long periods. As already reported in current literature both diets have already demonstrated an extremely positive impact on weight control and metabolic disease, without reporting no major side effects. In any case the diet prescription must be performed and monitored by qualified expert medical operators, that can follow accurately the patient’s health.

Please check and adjust the "Reference list" based on the regulations of reference list of journal. (Titles, doi, the name of journal and ... )

We adjusted the "Reference list" based on the regulations of reference list of journal.

Reviewer 4 Report

The current research evaluates in a very standardised methods the impact of weight loss per se, respectively comparison of 2 types of diet, on the hormonal millieu of PCOS cases, respectively on the performance of IVF procedures. 

I congratulate the authors for the idea and the design. However I have some comments:

- did you have any controls, PCOS cases without weight loss when looking into the IVF performance/complications such as OHSS.

- when evaluating the impact of weight loss on hormonal /ovarian functional parameters, an analysis regarding the correlation between the impact of weight loss, which of the parameters are independent in evolution   ( LH, FSM ratio,... ) when looking into body mass changes, ovarian function parameter changes, and which parameters are dependent. 

- can you identify, by using ROC curve analysis , the needed values for weight loss decrease, respectively hormonal balance changes, needed for a better ovarian response to IVF?

- you stated the in cases of amenorhea, the hormonal assay were made randomly. How come did you use, for the entire patient group mean LF,FSH, ... values, when oligomenorheic and amenorheic patients are different categories? 

Author Response

Thanks for congratulations and comments,

Did you have any controls, PCOS cases without weight loss when looking into the IVF performance/complications such as OHSS.

- We didn’t have a control group because overweight patients suffering from PCOS and insulin resistant, if not treated with a nutritional approach, must take oral hypoglycemics. Patients with BMI > 30, per our UOC policy, cannot access the IVF process.

When evaluating the impact of weight loss on hormonal /ovarian functional parameters, an analysis regarding the correlation between the impact of weight loss, which of the parameters are independent in evolution   ( LH, FSM ratio,... ) when looking into body mass changes, ovarian function parameter changes, and which parameters are dependent.

- The normalization of hormonal parameters (FSH, LH, FSH/LH ratio...) is a consequence of the recovery of ovarian function (ovulatory cycles). Metabolic parameters (cholesterol, triglycerides...) improve as a direct effect of a healthier eating style. At the same time, we wanted to emphasize that weight loss is the key moment for to break the vicious circle and leads to an improvement of metabolic and ovarian function.

Can you identify, by using ROC curve analysis , the needed values for weight loss decrease, respectively hormonal balance changes, needed for a better ovarian response to IVF?

The ROC curve is a statistical technique that measures the accuracy of a diagnostic test along the entire range of possible values. ROC curves are graphical schemes for binary classifiers. In our case, the variables under consideration as a function of weight loss do not represent parameters that can be defined through predetermined cut-offs, which would represent the limit between "positive" and "negative." This does not make them representable through a binary system, which is necessary for the development of an ROC test. The only parameter that can be represented by a dichotomous variable is the risk of developing OHSS. However, we are talking about risk and not actual disease, this does not allow us to define false positives and negatives, and even if it were possible, it would be beyond the scope of this paper.

You stated the in cases of amenorhea, the hormonal assay were made randomly. How come did you use, for the entire patient group mean LF,FSH, ... values, when oligomenorheic and amenorheic patients are different categories?

- The inversion of the FSH/LH ratio is a typical characteristic of patients suffering from PCOS as it is linked to the condition of anovulation or pauciovulation. So this is among the most significant parameters in these patients, both with amenorrhea and oligomenorrhea. The dosage of FSH and LH occurs within the 5th day of the menstrual flow, therefore in patients with amenorrhea the dosage cannot necessarily be carried out in relation to the menstrual cycle, in this sense it is carried out randomly.

Thanks for your consideration.